# Inequalities in the prevalence of cardiovascular disease risk factors in Brazilian slum populations: A cross-sectional study

Jasper J. L. Chan[1]*, Linh Tran-Nhu[2], Charlie F. M. Pitcairn[1,3], Anthony A. Laverty[4], Matías Mrejen[5], Julia M. Pescarini[6,7], Thomas V. Hone[4]

1 Imperial College School of Public Health, Imperial College London, London, United Kingdom, 2 Division of Biosciences, University College London, London, United Kingdom, 3 Faculty of Public Health and Policy, London School of Hygiene and Tropical Medicine, London, United Kingdom, 4 Public Health Policy Evaluation Unit, School of Public Health, Imperial College London, London, United Kingdom, 5 Instituto de Estudos para Políticas de Saúde (IEPS), São Paulo, SP, Brazil, 6 Centro de Integração de Dados e Conhecimentos para Saúde (Cidacs), Fundação Oswaldo Cruz, Salvador, Brazil, 7 Faculty of Epidemiology & Population Health, London School of Hygiene & Tropical Medicine, London, United Kingdom

* jasper.chan17@imperial.ac.uk

## Abstract

### Background

Social and environmental risk factors in informal settlements and slums may contribute to increased risk of cardiovascular disease (CVD). This study assesses the socioeconomic inequalities in CVD risk factors in Brazil comparing slum and non-slum populations.

### Methods

Responses from 94,114 individuals from the 2019 Brazilian National Health Survey were analysed. The United Nations Human Settlements Programme definition of a slum was used to identify slum inhabitants. Six behavioural risk factors, four metabolic risk factors and doctor-diagnosed CVD were analysed using Poisson regression models adjusting for socio-economic characteristics.

### Results

Compared to urban non-slum inhabitants, slum inhabitants were more likely to: have low (less than five days per week) consumption of fruits (APR: 1.04, 95%CI 1.01–1.07) or vegetables (APR: 1.08, 95%CI 1.05–1.12); drink four or more alcoholic drinks per day (APR: 1.05, 95%CI 1.03–1.06); and be physically active less than 150 minutes per week (APR: 1.03, 95%CI 1.01–1.04). There were no differences in the likelihoods of doctor-diagnosed metabolic risk factors or CVD between the two groups in adjusted models. There was a higher likelihood of behavioural and metabolic risk factors among those with lower education, with lower incomes, and the non-White population.

 

**Data Availability Statement:** All data is from the 2019 Brazilian Health Survey - Pesquisa Nacional de Saúde (PNS). The PNS is publicly available from

the Brazilian Institute of Geography and Statistics (IBGE): https://www.ibge.gov.br/estatisticas/sociais/saude/9160-pesquisa-nacional-de-saude.html?=&t=downloads.

**Funding:** The authors received no specific funding for this work.

**Competing interests:** The authors have declared that no competing interests exist.

## Conclusions

Brazilians living in slums are at higher risk of behavioural risk factors for CVD, suggesting local environments might impact access to and uptake of healthy behaviours.

## Introduction

Development and urbanisation have fundamentally affected demographics and lifestyles globally–especially in low- and middle- income countries (LMICs) [1]. Consequently, chronic non-communicable diseases (NCDs) are now a major health challenge [2]. Cardiovascular disease (CVD) is the leading contributor to global NCD mortality and morbidity–the number one cause of deaths globally and responsible for almost 18 million deaths annually [3]. Nearly 80% of all CVD deaths occur in LMICs, where the annual cost of management and treatment exceeds the total health expenditure per capita [1,4].

CVD stems from a combination of interlinked risk factors [5]. This includes non-modifiable risk factors, such as sex, age, and race [5], however, modifiable behavioural and metabolic risk factors play a larger role [6,7]. Behavioural risk factors include an unhealthy diet (e.g. lacking in fruits and vegetables), physical inactivity, tobacco use and alcohol consumption [5,6,8,9]. These lifestyle risk factors drive increases in intermediate metabolic risk factors such as obesity, hypertension, diabetes and high cholesterol [5,6,8]. Unaddressed, these risk factors increase the risk of CVD events such as heart disease or stroke [10–12].

Individuals living in slums and informal settlements are a growing population–nearly one billion people worldwide now live in slums, a figure expected to double by 2030 [13,14]. Slums have grown sizeably in LMICs due to inadequate urban planning and rapid urbanisation [13,15]. Slums are characterised by poverty, overcrowding, inadequate housing and limited access to safe water and sanitation [15–17]. There is also evidence that individuals living within them are at greater risk of CVD and have higher rates of behavioural CVD risk factors compared to populations in formal urban or rural settlements [18–21]. However, the health profiles and risk factors of slum populations are infrequently studied [13,22,23].

The wider social determinants of health are important determinants of CVD and associated risk factors [24]. Individuals of lower socioeconomic status (SES) experience increased morbidity and mortality on average [25]. Lower income, lower educational attainment, unemployment, and living in disadvantaged areas is associated with higher rates of CVD [26–28]. In many countries, the high costs of treating and managing CVDs can drive individuals and families into poverty, worsening SES, and passing disadvantage from one generation to the next [1,4,29,30]. Much of the literature on CVD and SES inequalities comes from high-income countries [28,31]. However, it is important to generate evidence from LMICs where globalisation has rapidly changed lifestyles in recent decades–there are large inequalities and sizeable slum populations, and access to high-quality healthcare is not universal [14,32].

Brazil is a country with large health and social inequalities [22,33]. Like many other countries, an aging population, obesity, unhealthy diets, and physical inactivity have contributed to a rising burden of CVDs–responsible for 28% of all NCD deaths in Brazil in 2016 [12,34,35]. Brazil is the country with the greatest degree of wealth inequality in South America where the majority of wealth belongs to the top one percent [36,37], and over a quarter of the population live in poverty [38]. Additionally, over 16% of Brazil's urban population reside in slum settlements (*favelas*) [21,22,39]. However, few studies have examined the CVD risk factors and prevalence in these settings [17,22]. National studies demonstrate socioeconomic inequalities

in NCD risk factors and health behaviours [12,40]. For example, rural populations have lower average incomes, are more exposed to infectious diseases, and face greater barriers to accessing healthcare. However, there are no country-wide studies comparing behavioural and metabolic CVD risk factors between rural, slum and non-slum urban inhabitants. Better knowledge of slum population health and their CVD risk factors is important for making progress towards the United Nations Sustainable Development Goals (SDGs) which include targets for reducing premature NCDs deaths (SDG 3), reducing inequality within countries (SDG 10), and upgrading slums (SDG 11) [41].

This study assesses the prevalence of behavioural and metabolic CVD risk factors across socioeconomic and demographic groups in Brazil in 2019. It investigates the associations between socioeconomic inequalities and the prevalence of CVD and its risk factors, specifically examining the inequalities between slum and non-slum inhabitants.

## Methods

### Ethics statement

Ethics approval for the PNS 2019 itself was approved by Brazil's National Research Ethics Committee/National Health Council (opinion no. 3.529.376) with informed consent obtained for all respondents. Ethics approval for this specific study was not required as it utilised anonymised data in the public domain.

### Study design

This is a cross-sectional analysis of the 2019 Brazilian National Health Survey–*Pesquisa Nacional de Saúde* (PNS).

### Data source

Responses from the 2019 PNS were obtained from the website of the Brazilian Institute of Geography and Statistics (IBGE). The PNS is a country-wide household-based survey conducted by IBGE and the Ministry of Health. It collects information on the health, healthcare access and usage, and lifestyles of the Brazilian population [42]. A three-stage probability sample selection process was used. First, primary sampling units (PSU) composed of one or more census tracts were selected by random sampling. Second, 12–18 households were randomly selected from each PSU. Third and finally, one individual aged 15 years or older was selected from each household for a computer-assisted personal interview. Interviewed participants answered questionnaires on housing, socioeconomic and demographic characteristics, health status and behaviours. The PNS is publicly available from IBGE: https://www.ibge.gov.br/estatisticas/sociais/saude/9160-pesquisa-nacional-de-saude.html?=&t=downloads.

A total of 293,725 individuals from 108,457 households were surveyed across Brazil. Of these, 94,114 individuals answered the individual questionnaires and were included in this study. Responses were weighted to be representative of the national Brazilian population [43].

### Outcomes

Eleven binary outcomes, including ten modifiable risk factors and doctor-diagnosed CVD, were derived as outcomes in this analysis. The risk factors were divided into six behavioural and four metabolic ones. Risk factors were chosen based on evidence of their association with CVD [2,5–8]. The six behavioural risk factors were self-reported: low weekly consumption of vegetables (<5 days per week); low weekly consumption of fruits (<5 days per week); high weekly consumption of red meat (≥3 days per week); currently smoking tobacco; heavy

consumption of alcoholic drinks ($\geq$4 drinks per day); and low weekly physical activity ($<$150 minutes per week). The categorisation for alcohol consumption was based on evidence from the United States National Institute on Alcohol Abuse and Alcoholism, which defined heavy alcohol consumption as four or more drinks per day [44–46]. For physical activity, the categorisation was based on the World Health Organization (WHO) recommendation of at least 150 minutes per week of moderate-intensity physical activity [47].

Four metabolic risk factor variables were included: overweight (body mass index (BMI) over 25kg/m$^2$) [48]; doctor-diagnosed hypertension; doctor-diagnosed diabetes; and doctor-diagnosed high cholesterol. Both height and weight were measured at the interview to determine BMI. The doctor-diagnosed conditions were determined in the interview as: "Has a doctor ever diagnosed you with. . .". Doctor-diagnosed CVD was also included as an outcome and was defined as an individual who reported ever having been diagnosed with either heart disease or stroke.

## Variables of interest

The United Nations Human Settlements Programme (UN-Habitat) definition of slums was used to identify respondents living in slums and has been used in similar studies in Brazil [49,50]. It defines slums as an environment where inhabitants lack access to one or more of these conditions: improved water source, improved sanitation facilities, sufficient living area, housing durability and security of tenure (Table 1) [51]. Data on a lack of security of tenure was not available and not included. In this study, if an individual met any of the criteria and were living in an urban area, they were considered slum residents. Those who did not meet the criteria were categorised as either urban non-slum or rural residents.

Ten socioeconomic and demographic variables were derived from the survey: area (slum, urban non-slum, rural); sex (male, female); age (15–29 years, 30–39, 40–49, 50–59, 60–69, 70 or older); self-reported race/ethnicity (White, Black, *Pardo*/mixed (Brown), other (Asian, Indigenous or not reported)); highest level of education (illiterate, elementary education, high school education, higher education); currently in the workforce (yes, no); household income per capita ($\leq$ half minimum wage, $>$ half but $\leq$1 minimum wage, $>$1 but $\leq$2 minimum wage, $>$2 minimum wage); household registered to the Family Health Strategy (FHS) (yes, no, do not know); private healthcare (yes, no); and the last time a doctor was consulted ($\leq$1 year, $>$1 but $\leq$2 years, $>$2 years or never consulted). Those who identified as Asian or Indigenous were grouped together with those who ignored the question due to small numbers. In 2020, the annual minimum wage in Brazil was USD 5,198.40 after adjusting for purchasing power [53]. The FHS is Brazil's main primary healthcare program and consists of primary care services delivered through multidisciplinary teams assigned to a certain area, comprising of approximately 3,500 individuals [54,55].

## Data analysis

All analyses accounted for clustered sampling of the survey and weighted for non-response. Descriptive data analysis explored the prevalence of the socioeconomic characteristics, behavioural risk factors, metabolic risk factors and CVD outcomes in the study population. Weighted prevalence estimates (%) and 95% confidence intervals (95%CIs) are reported. Prevalence estimates were explored across slum, urban non-slum and rural populations.

Adjusted Poisson regression models were used to analyse the associations between demographic and socioeconomic variables with different CVD outcomes. Poisson regression models were chosen due to the high prevalence of many risk factors and to obtain prevalence ratios which are more interpretable. Models were repeated for each of the eleven outcomes, and all

**Table 1. Slum residence definition.**

| UN-Habitat definition | Variable | Full PNS survey question | Responses considered for slums |
|---|---|---|---|
| Lack of access to improved water source | Water supply | What is the main form of water supply for this household? | Shallow, water table or *cacimba* |
| | | | Source or spring |
| | | | Another |
| | Water distribution network | Is this household connected to the general water distribution network? | No |
| | Arriving water | How does the water used in this household arrive? | Not piped |
| Lack of access to improved sanitation | Bathroom or sanitation room | How many bathrooms or sanitation rooms for residents to use are located on the ground or property? | None |
| | Dejections | Do residents use a toilet or hole located on the ground or property for dejections? | No |
| | Drainage | Where does the sullage from the bathroom, toilet or hole go? | Rudimentary pit |
| | | | Ditch |
| | | | River, lake, stream or sea |
| | | | Another |
| Lack of sufficient living area | Number of rooms | How many rooms does this household have? | ≥3 people per room* |
| | Number of household members | Number of household members | |
| Lack of housing durability | Wall material | What is the predominant material on the external walls of the household? | Uncoated rammed earth |
| | | | Harnessed wood |
| | Roof material | What is the predominant material on the roof of the household? | Zinc, aluminium or sheet metal |
| | Floor material | What is the predominant material on the floor of the household? | Earth |

This was based on the United Nations Human Settlements Programme (UN-Habitat) definition of slum households with indicators mapped to questions from the 2019 Brazilian National Health Survey, *Pesquisa Nacional de Saúde* (PNS).

*Overcrowding is defined by UN-Habitat as three or more people per room [52].

models were adjusted for location of residence (urban slum, urban non-slum, rural), sex, age, race/ethnicity, educational level, employment status, and household income per capita. Collinearity of variables were tested and found to be unproblematic as all variance inflation factors were less than 2. Adjusted prevalence ratios (APRs) with their respective 95%CIs and *p*-value are reported. The unadjusted regression models can be found in the supporting information (S1 and S2 Tables). In supplementary analyses, differential associations between slum residence and the eleven outcomes were explored across the regions of Brazil. Models were repeated with interactions between the variable for location of residence (urban slum, urban

**Table 2. Distribution of study population's area of residence.**

| | n | Weighted % | 95%CI |
|---|---|---|---|
| **Area** | | | |
| Slum | 16,627 | 14.4 | 13.7–15.0 |
| Urban non-slum | 56,082 | 71.6 | 70.8–72.4 |
| Rural | 21,405 | 14.1 | 13.7–14.5 |

Source: Brazilian National Health Survey *Pesquisa Nacional de Saúde* (PNS) 2019. Shows numbers of individuals (n) and the weighted percentage (%) prevalence estimates. 95%CI—95% confidence interval. Percentages may not equal 100% due to rounding.

non-slum, rural) and region of residence (North, Northeast, Southeast, South and Central west). Prediction prevalence estimates from interacted Poisson regression models were plotted (S1 Fig).

Stata/IC 16.1 was used to analyse the data.

## Results

The responses from 94,114 individuals were included. According to the UN-Habitat definition, 14.4% (95%CI 13.7–15.0) of the Brazilian population are slum residents (Table 2).

The self-reported prevalence and distribution of the socioeconomic and demographic characteristics is shown in Table 3. The average age of the participants was 46.2 years, with individuals living in slums generally younger. Slums had the highest prevalence of participants identifying as Black (12.1%) or *Pardo*/mixed (57.0%). In terms of education, nearly half of slum populations reported having high school (36.9%) or higher education (11.3%) which was lower than urban non-slums populations but higher than rural populations. Rural populations had the greatest proportion of illiteracy (13.3%) and over half of the population had elementary education (56.9%) but did not progress further with only 4.3% having higher education. Rural populations also had the greatest proportion of unemployment (44.2%) compared to slums and urban non-slums (37.6% and 33.7% respectively). Over two thirds of the population in slums (69.4%) earned the minimum wage or less, compared to 78.5% in rural areas and 43.8% in urban non-slum areas. In slums, 70.3% of households were registered with the FHS compared 78.6% in rural areas and 56.7% in urban non-slums. Conversely, urban non-slums had the highest access to private healthcare (33.3%) compared to 12.8% of respondents in slums and 6.0% of rural respondents. The proportion of individuals who had a doctor consultation within the last year was highest in urban non-slums (82.6% of respondents) compared to 75.2% in slums and 70.7% in rural areas.

The prevalence of modifiable CVD risk factors was examined by area of residence (Fig 1). Compared to urban non-slum populations, individuals in slums and rural areas had a higher (unadjusted) prevalence of: low fruits consumption (61.7% (slum) and 64.9% (rural), compared to 52.8% (urban non-slum)); low vegetables consumption (53.0%, 57.3%, and 42.2%); smoking (12.8%, 12.9%, and 11.9%); heavy alcohol consumption (82.9%, 81.0%, and 74.4%); and physical inactivity (76.1%, 83.1%, and 70.1%). Only for high red meat consumption, was the prevalence highest in urban non-slums (65.8% (urban non-slum), 60.0% (slum), and 62.8% (rural)). For metabolic CVD risk factors (Fig 2), 55.4% of slum residents and 57.3% of urban non-slum residents were overweight compared to rural residents (49.1%). For doctor-diagnosed hypertension, diabetes, high cholesterol and CVD, the prevalence was higher in urban environments than rural areas.

Table 4 shows results from adjusted Poisson regression models on behavioural CVD risk factors. The unadjusted regression can be found in S1 Table. In fully-adjusted models and compared to urban non-slum populations, residents of slums had: 8.3% higher prevalence of having low vegetables consumption (APR: 1.083, 95%CI 1.048–1.119); 3.7% higher prevalence of having low fruits consumption (APR: 1.037 95%CI 1.010–1.065); 4.5% higher prevalence of having heavy alcohol consumption (APR: 1.045, 95%CI 1.028–1.062); and 2.5% higher prevalence of being physically inactive (APR: 1.025, 95%CI 1.007–1.044). Individuals from rural areas also had a higher prevalence of behavioural risk factors compared to urban non-slums. However, both rural and slum inhabitants had a lower prevalence of smoking (APR: 0.806, 95%CI 0.742–0.875 and APR: 0.920, 95%CI 0.847–0.999 respectively). Regarding other SES factors, women had a higher prevalence of high alcohol consumption (APR: 1.046, 95%CI 1.032–1.061) and being physically inactive (APR: 1.079, 95%CI 1.062–1.096), but had a lower

**Table 3. Socioeconomic and demographic characteristics of study population by area of residence.**

| | n | Slum<br>Weighted % (95%CI) | Urban non-slum<br>Weighted % (95%CI) | Rural<br>Weighted % (95%CI) |
|---|---|---|---|---|
| **Sex** | | | | |
| Male | 44,752 | 46.4 (45.2–47.7) | 45.9 (45.2–46.7) | 53.5 (52.2–54.8) |
| Female | 49,362 | 53.6 (52.3–54.8) | 54.1 (53.3–54.8) | 46.5 (45.2–47.8) |
| **Age** | | | | |
| 15–29 | 18,648 | 28.8 (27.5–30.2) | 25.6 (24.8–26.4) | 27.9 (26.6–29.2) |
| 30–39 | 18,904 | 20.2 (19.2–21.2) | 19.9 (19.3–20.5) | 19.2 (18.4–20.1) |
| 40–49 | 17,282 | 17.6 (16.6–18.5) | 17.2 (16.6–17.7) | 16.8 (16.2–17.5) |
| 50–59 | 16,136 | 16.2 (15.4–17.2) | 16.4 (15.9–16.9) | 15.0 (14.3–15.7) |
| 60–69 | 12,799 | 9.4 (8.8–10.1) | 11.9 (11.5–12.4) | 11.4 (10.7–12.1) |
| 70+ | 10,345 | 7.8 (7.2–8.4) | 9.0 (8.6–9.5) | 9.7 (9.2–10.3) |
| **Race/ethnicity** | | | | |
| White | 34,320 | 29.7 (28.2–31.2) | 47.7 (46.8–48.7) | 32.1 (30.8–33.5) |
| Black | 10,730 | 12.1 (11.3–13.0) | 11.3 (10.8–11.9) | 10.9 (10.1–11.9) |
| *Pardo*/mixed (Brown) | 47,660 | 57.0 (55.5–58.6) | 39.3 (38.4–40.1) | 56.0 (54.6–57.5) |
| Other | 1,404 | 1.2 (1.0–1.6) | 1.7 (1.5–1.9) | 0.9 (0.7–1.1) |
| **Education level (complete or incomplete)** | | | | |
| Illiterate | 7,870 | 8.5 (7.7–9.4) | 3.8 (3.6–4.1) | 13.3 (12.7–14.1) |
| Elementary education | 36,829 | 43.3 (41.8–44.7) | 32.2 (31.4–33.0) | 56.9 (55.6–58.1) |
| High school education | 31,128 | 36.9 (35.6–38.3) | 39.4 (38.6–40.1) | 25.5 (24.3–26.7) |
| Higher education | 18,287 | 11.3 (10.5–12.1) | 24.6 (23.6–25.5) | 4.3 (3.9–4.7) |
| **Currently employed** | | | | |
| Yes | 59,172 | 62.4 (61.2–63.6) | 66.3 (65.6–67.1) | 55.8 (54.7–57.0) |
| No | 34,942 | 37.6 (36.4–38.8) | 33.7 (33.0–34.4) | 44.2 (43.0–45.3) |
| **Household income per capita** | | | | |
| ≤ half minimum wage | 24,592 | 34.3 (32.8–35.8) | 16.3 (15.6–17.0) | 46.4 (45.2–47.7) |
| > half but ≤ 1 minimum wage | 27,417 | 35.1 (33.9–36.3) | 27.5 (26.8–28.3) | 32.0 (30.9–33.1) |
| > 1 but ≤ 2 minimum wage | 23,248 | 21.6 (20.4–22.7) | 31.0 (30.3–31.8) | 16.0 (15.1–16.9) |
| > 2 minimum wage | 18,833 | 9.1 (8.4–9.9) | 25.2 (24.2–26.2) | 5.6 (5.2–6.1) |
| **Household registered to the Family Health Strategy** | | | | |
| Yes | 59,358 | 70.3 (68.4–72.2) | 56.7 (55.3–58.2) | 78.6 (76.9–80.2) |
| No | 23,424 | 19.7 (18.1–21.4) | 31.1 (29.8–32.4) | 13.0 (11.8–14.4) |
| Do not know | 11,332 | 10.0 (9.1–11.0) | 12.2 (11.5–12.9) | 8.4 (7.4–9.4) |
| **Private healthcare** | | | | |
| Yes | 21,333 | 12.8 (11.9–13.9) | 33.3 (32.3–34.4) | 6.0 (5.3–6.6) |
| No | 72,781 | 87.2 (86.1–88.1) | 66.7 (65.6–67.7) | 94.0 (93.4–94.7) |
| **The last time a doctor was consulted** | | | | |
| ≤1 year | 73,561 | 75.2 (74.1–76.4) | 82.6 (81.9–83.2) | 70.7 (69.4–72.0) |
| >1 but ≤2 years | 9,885 | 11.1 (10.4–11.9) | 9.1 (8.7–9.6) | 12.1 (11.3–12.9) |
| >2 years or never consulted | 10,668 | 13.6 (12.6–14.6) | 8.3 (7.9–8.7) | 17.2 (16.1–18.3) |

Source: Brazilian National Health Survey *Pesquisa Nacional de Saúde* (PNS) 2019. Shows numbers of individuals (n) and the weighted percentage (%) prevalence estimates. 95%CI—95% confidence interval. Percentages may not equal 100% due to rounding.

prevalence of low fruits and vegetables consumption (APR: 0.844, 95% CI 0.827–0.861 and APR: 0.835, 95%CI 0.814–0.856 respectively), high red meat consumption (APR: 0.855, 95%CI 0.841–0.870), and smoking (APR: 0.637, 95%CI 0.599–0.677). There was a higher prevalence

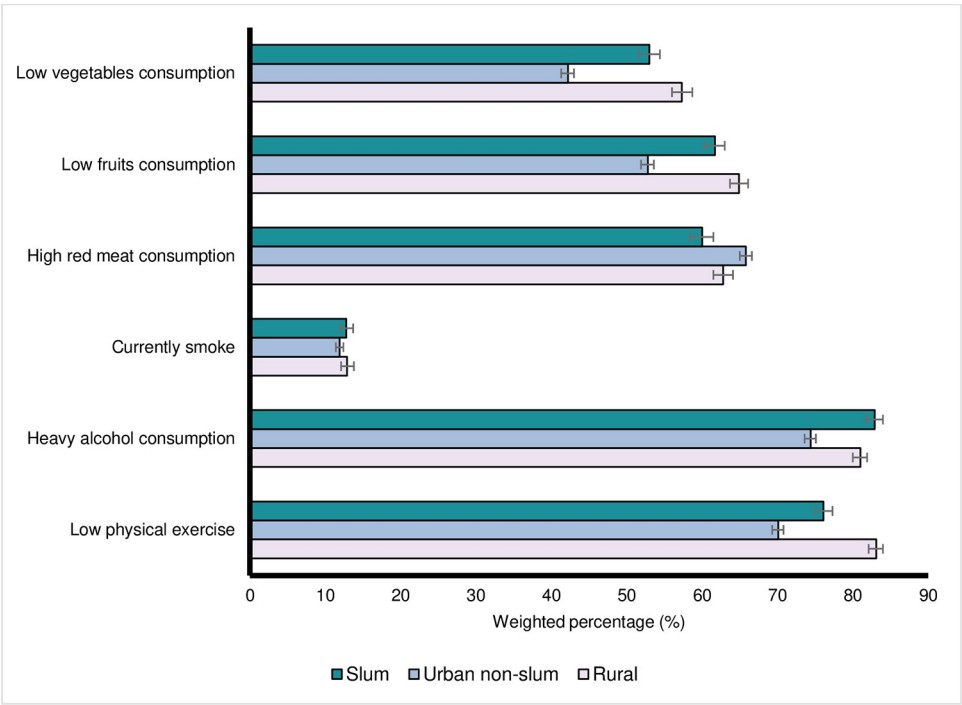

**Fig 1. Weighted prevalence of self-reported cardiovascular disease (CVD) behavioural risk factors by area of residence.** Source: Brazilian National Health Survey *Pesquisa Nacional de Saúde* (PNS) 2019. Low Vegetables Consumption–Vegetables consumption <5 days per week; Low Fruits Consumption–Fruits consumption <5 days per week; High Red Meat Consumption–Red meat consumption ≥3 days per week; Currently Smoke–Currently smoke tobacco; Heavy Alcohol Consumption–Alcohol consumption ≥4 per day; Low Physical Exercise–Exercise <150 minutes per week.

of low fruits and vegetables consumption and heavy alcohol use among the non-white population. Individuals with higher levels of education had a lower prevalence of low levels of fruits and vegetables consumption, smoking tobacco, consuming high amounts of alcohol, or being physically inactive. Generally, as household income increased, the prevalence of behavioural CVD risk factors decreased.

Table 5 shows the adjusted Poisson regression results for metabolic risk factors for CVD and doctor-diagnosed CVD. The unadjusted regression can be found in S2 Table. In fully-adjusted models, there was no statistically significant difference in all self-reported doctor-diagnosed conditions (overweight, hypertension, diabetes, high cholesterol and CVD) between slums and urban non-slums. However, individuals living in rural areas had a lower prevalence of all metabolic risk factors and CVD. Females had a higher prevalence of hypertension (APR: 1.262, 95%CI 1.218–1.309), diabetes (APR: 1.097, 95%CI 1.013–1.189) and high cholesterol (APR: 1.413, 95%CI 1.334–1.497). The prevalence of having hypertension was significantly higher in the non-white population. Notably, there were no significant differences in the prevalence of high cholesterol and CVD across education levels. Across education levels, illiterate individuals had a lower prevalence of being overweight, whilst increasing educational attainment was associated with a lower prevalence of hypertension and diabetes. Unemployed individuals had a higher prevalence of hypertension, diabetes, high cholesterol, and CVD yet had a lower prevalence of being overweight. There were no clear patterns by household income, except for high cholesterol which was associated with increasing income.

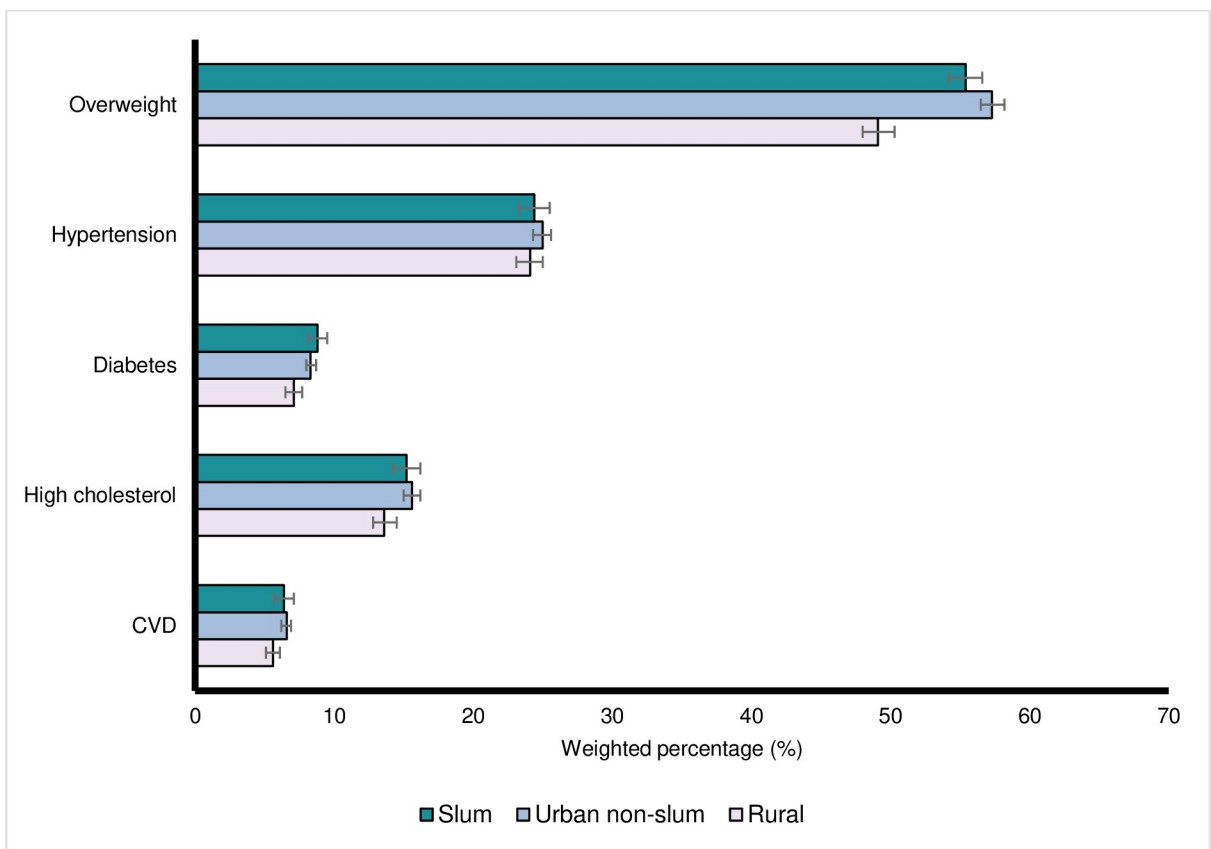

**Fig 2. Weighted prevalence of self-reported cardiovascular disease (CVD) metabolic risk factors and doctor-diagnosed CVD by area of residence.** Source: Brazilian National Health Survey *Pesquisa Nacional de Saúde* (PNS) 2019. CVD–doctor-diagnosed with either heart disease or stroke.

When examining the results by region, there were quite large differences across Brazil (S1 Fig). Compared the wealthier South, Southeast and Central west regions, the North and Northeast regions had generally lower levels of vegetable consumption, lower red meat consumption, lower smoking, and higher excessive drinking. There was a general trend of lower hypertension and CVD.

By region and area of residence, many confidence intervals were overlapping or showed similar trends to the overall pooled regressions. However, there were a few notable findings. Compared to non-slum urban and rural populations, slum populations in the North had higher vegetable consumption, whilst slum populations in the Northeast and South-east had lower red meat consumption. Over 85% of rural populations in the central west had high red meat consumption compared to ~50% of rural northern populations. Excessive drinking varied from ~87% of non-slum urban populations in the North to ~66% of rural populations in the South. Across the regions, rural populations were more likely to have low physical activity, and in the North and Northeast rural populations were less likely to have a BMI over 25.

## Discussion

This study found that, compared to non-slum urban populations, Brazilian slum and rural populations had a greater likelihood of having many behavioural CVD risk factors (including low fruits and vegetables consumption, heavy alcohol consumption (only slum) and physical

**Table 4. Poisson regression models on the association between slums and behavioural cardiovascular disease risk factors.**

| | Low Vegetables Consumption | Low Fruits Consumption | High Red Meat Consumption | Currently Smoke | Heavy Alcohol Consumption | Low Physical Exercise |
|---|---|---|---|---|---|---|
| | APR | APR | APR | APR | APR | APR |
| | 95%CI | 95%CI | 95%CI | 95%CI | 95%CI | 95%CI |
| **Area** | | | | | | |
| Non-slum urban (Ref) | 1 | 1 | 1 | 1 | 1 | 1 |
| Slum | 1.083*** | 1.037** | 0.962** | 0.920* | 1.045*** | 1.025** |
| | 1.048–1.119 | 1.010–1.065 | 0.934–0.990 | 0.847–0.999 | 1.028–1.062 | 1.007–1.044 |
| Rural | 1.100*** | 1.037** | 1.036* | 0.806*** | 0.996 | 1.077*** |
| | 1.067–1.135 | 1.012–1.062 | 1.009–1.063 | 0.742–0.875 | 0.981–1.011 | 1.060–1.095 |
| **Sex** | | | | | | |
| Male (Ref) | 1 | 1 | 1 | 1 | 1 | 1 |
| Female | 0.835*** | 0.844*** | 0.855*** | 0.637*** | 1.046*** | 1.079*** |
| | 0.814–0.856 | 0.827–0.861 | 0.841–0.870 | 0.599–0.677 | 1.032–1.061 | 1.062–1.096 |
| **Age** | | | | | | |
| 15–29 (Ref) | 1 | 1 | 1 | 1 | 1 | 1 |
| 30–39 | 0.830*** | 0.920*** | 0.980 | 1.225*** | 1.007 | 1.101*** |
| | 0.799–0.863 | 0.893–0.947 | 0.956–1.004 | 1.101–1.364 | 0.984–1.030 | 1.073–1.131 |
| 40–49 | 0.753*** | 0.821*** | 0.936*** | 1.261*** | 0.983 | 1.120*** |
| | 0.724–0.783 | 0.795–0.848 | 0.912–0.961 | 1.129–1.407 | 0.960–1.007 | 1.091–1.150 |
| 50–59 | 0.696*** | 0.741*** | 0.865*** | 1.492*** | 0.973* | 1.155*** |
| | 0.667–0.727 | 0.715–0.768 | 0.840–0.890 | 1.335–1.666 | 0.951–0.996 | 1.125–1.187 |
| 60–69 | 0.668*** | 0.651*** | 0.853*** | 1.364*** | 0.977* | 1.196*** |
| | 0.638–0.701 | 0.624–0.679 | 0.825–0.881 | 1.214–1.533 | 0.953–1.001 | 1.161–1.231 |
| 70+ | 0.593*** | 0.544*** | 0.851*** | 0.729*** | 1.007 | 1.293*** |
| | 0.561–0.627 | 0.517–0.572 | 0.818–0.886 | 0.629–0.845 | 0.980–1.034 | 1.254–1.334 |
| **Race/ethnicity** | | | | | | |
| White (Ref) | 1 | 1 | 1 | 1 | 1 | 1 |
| Black | 1.129*** | 1.057** | 0.965* | 1.007 | 1.059*** | 0.976* |
| | 1.084–1.176 | 1.023–1.093 | 0.936–0.994 | 0.909–1.115 | 1.033–1.084 | 0.954–0.999 |
| *Pardo*/mixed (Brown) | 1.096*** | 1.026* | 0.959*** | 1.009 | 1.071*** | 0.975** |
| | 1.063–1.131 | 1.002–1.050 | 0.941–0.977 | 0.941–1.082 | 1.055–1.087 | 0.958–0.991 |
| Other | 1.020 | 1.007 | 0.866** | 1.033 | 1.070 | 1.030 |
| | 0.891–1.167 | 0.902–1.123 | 0.783–0.958 | 0.808–1.320 | 0.981–1.168 | 0.964–1.101 |
| **Education level** | | | | | | |
| Illiterate (Ref) | 1 | 1 | 1 | 1 | 1 | 1 |
| Elementary education | 0.797*** | 0.818*** | 1.107*** | 0.810*** | 0.941*** | 0.976* |
| | 0.764–0.831 | 0.790–0.847 | 1.055–1.161 | 0.735–0.892 | 0.925–0.956 | 0.957–0.996 |
| High school education | 0.761*** | 0.762*** | 1.122*** | 0.495*** | 0.917*** | 0.884*** |
| | 0.727–0.797 | 0.734–0.792 | 1.068–1.179 | 0.441–0.557 | 0.898–0.936 | 0.862–0.906 |
| Higher education | 0.667*** | 0.651*** | 1.117*** | 0.400*** | 0.828*** | 0.752*** |
| | 0.628–0.709 | 0.619–0.684 | 1.060–1.178 | 0.345–0.465 | 0.804–0.852 | 0.727–0.778 |
| **Currently employed** | | | | | | |
| Yes (Ref) | 1 | 1 | 1 | 1 | 1 | 1 |
| No | 1.039** | 0.980 | 0.941*** | 0.798*** | 1.095*** | 0.946*** |
| | 1.010–1.068 | 0.957–1.004 | 0.920–0.963 | 0.740–0.859 | 1.078–1.113 | 0.928–0.964 |
| **Household income per capita** | | | | | | |
| ≤ half minimum wage (Ref) | 1 | 1 | 1 | 1 | 1 | 1 |

*(Continued)*

**Table 4.** (Continued)

| | Low Vegetables Consumption | Low Fruits Consumption | High Red Meat Consumption | Currently Smoke | Heavy Alcohol Consumption | Low Physical Exercise |
|---|---|---|---|---|---|---|
| | APR | APR | APR | APR | APR | APR |
| | 95%CI | 95%CI | 95%CI | 95%CI | 95%CI | 95%CI |
| > half but ≤ 1 minimum wage | 0.892*** | 0.929*** | 1.165*** | 0.911* | 0.990 | 0.951*** |
| | 0.865–0.919 | 0.906–0.953 | 1.134–1.197 | 0.838–0.991 | 0.973–1.006 | 0.935–0.968 |
| > 1 but ≤ 2 minimum wage | 0.767*** | 0.846*** | 1.241*** | 0.832*** | 0.944*** | 0.917*** |
| | 0.739–0.796 | 0.821–0.871 | 1.205–1.278 | 0.757–0.914 | 0.926–0.963 | 0.898–0.937 |
| > 2 minimum wage | 0.649*** | 0.721*** | 1.264*** | 0.782*** | 0.851*** | 0.812*** |
| | 0.615–0.685 | 0.689–0.754 | 1.222–1.307 | 0.696–0.877 | 0.829–0.873 | 0.789–0.837 |
| Number of observations | 90,824 | 90,824 | 90,824 | 90,824 | 90,824 | 90,824 |

Source: Brazilian National Health Survey *Pesquisa Nacional de Saúde* (PNS) 2019. APR–Adjusted prevalence ratio; 95%CI– 95% Confidence interval; Ref–Reference category

*$p<0.05$

**$p<0.01$

***$p<0.001$

Low Vegetables Consumption–Vegetables consumption <5 days per week; Low Fruits Consumption–Fruits consumption <5 days per week; High Red Meat Consumption–Red meat consumption ≥3 days per week; Currently Smoke–Currently smoke tobacco; Heavy Alcohol Consumption–Alcohol consumption ≥4 per day; Low Physical Exercise–Exercise <150 minutes per week.

inactivity). Compared to non-slum urban populations, there were no differences in the likelihood of doctor-diagnosed metabolic risk factors or CVD for slum populations, however rural populations had a lower prevalence across all metabolic risk factors and CVD. Sizeable SES inequalities in CVD risk factors and doctor-diagnosed CVD exist in Brazil by sex, race, education and income.

The findings of an association between slum residence and higher rates of behavioural CVD risk factors (which persisted after adjusting for socioeconomic factors) is generally concordant with other studies and are likely to be partially explained by the slum environment. Studies from Kenyan and Nepalese slums found slum populations reported lower fruits and vegetables consumption, higher heavy alcohol consumption, and more physical inactivity [56,57]. Low fruits and vegetables consumption suggest barriers to accessing healthy foods in slums [58]. Poor transport infrastructure, limited public services and unsafe neighbourhoods in slums have been found to affect local food availability [17,59]. Violence can also deter local businesses and violent drug-related crime is a particular challenge in Brazilian *favelas* [60]. Violence and unsafe neighbourhoods can also deter exercise–concordant with evidence from Kenyan slums [58]. Furthermore, the high population density of slums limits space for exercise and gyms in slums are often unaffordable or unavailable [20]. In this study, slum dwellers were also more likely to have excessive alcohol consumption–similar to evidence from Indian slums [61]. In Brazil, the availability and advertisement of alcohol is widespread, with alcohol consumption increasing over the last decade [40]. Stress from poor quality housing and insecure neighbourhoods may contribute to increased alcohol consumption [62,63].

In adjusted models, there were no differences between slum dwellers and urban non-slum residents in the likelihood of being overweight or having doctor-diagnosed hypertension, diabetes, high cholesterol, or CVD. However, there was a significantly reduced prevalence in the rural population of these metabolic risk factors. This suggests socioeconomic factors, such as

**Table 5. Poisson regression models on the association between slums and metabolic cardiovascular disease risk factors.**

| | Overweight | Hypertension | Diabetes | High Cholesterol | CVD |
|---|---|---|---|---|---|
| | APR | APR | APR | APR | APR |
| | 95%CI | 95%CI | 95%CI | 95%CI | 95%CI |
| **Area** | | | | | |
| Non-slum urban (Ref) | 1 | 1 | 1 | 1 | 1 |
| Slum | 0.992 | 0.987 | 1.030 | 1.044 | 1.006 |
| | 0.966–1.019 | 0.941–1.034 | 0.942–1.127 | 0.969–1.125 | 0.891–1.136 |
| Rural | 0.890*** | 0.908*** | 0.731*** | 0.911* | 0.791*** |
| | 0.865–0.916 | 0.871–0.946 | 0.665–0.804 | 0.847–0.980 | 0.711–0.881 |
| **Sex** | | | | | |
| Male (Ref) | 1 | 1 | 1 | 1 | 1 |
| Female | 0.967** | 1.262*** | 1.097* | 1.413*** | 0.923 |
| | 0.947–0.988 | 1.218–1.309 | 1.013–1.189 | 1.334–1.497 | 0.846–1.007 |
| **Age** | | | | | |
| 15–29 (Ref) | 1 | 1 | 1 | 1 | 1 |
| 30–39 | 1.676*** | 3.026*** | 3.017*** | 1.682*** | 1.690** |
| | 1.610–1.745 | 2.624–3.489 | 2.215–4.110 | 1.421–1.991 | 1.233–2.315 |
| 40–49 | 1.784*** | 5.756*** | 5.557*** | 2.953*** | 2.866*** |
| | 1.710–1.861 | 4.984–6.647 | 4.171–7.405 | 2.527–3.451 | 2.159–3.804 |
| 50–59 | 1.765*** | 9.358*** | 11.40*** | 4.434*** | 5.266*** |
| | 1.686–1.847 | 8.248–10.62 | 8.556–15.18 | 3.808–5.164 | 4.029–6.883 |
| 60–69 | 1.797*** | 12.53*** | 16.00*** | 5.191*** | 7.234*** |
| | 1.715–1.883 | 11.06–14.20 | 12.06–21.23 | 4.447–6.061 | 5.502–9.511 |
| 70+ | 1.582*** | 14.30*** | 16.40*** | 4.446*** | 10.38*** |
| | 1.500–1.669 | 12.60–16.22 | 12.17–22.10 | 3.756–5.262 | 7.895–13.65 |
| **Race/ethnicity** | | | | | |
| White (Ref) | 1 | 1 | 1 | 1 | 1 |
| Black | 1.036* | 1.188*** | 1.120 | 0.951 | 0.996 |
| | 1.002–1.071 | 1.122–1.257 | 0.999–1.255 | 0.862–1.051 | 0.878–1.131 |
| *Pardo*/mixed (Brown) | 1.008 | 1.079*** | 1.069 | 0.980 | 0.954 |
| | 0.984–1.033 | 1.038–1.122 | 0.984–1.161 | 0.922–1.041 | 0.869–1.047 |
| Other | 0.932 | 1.143* | 1.445* | 1.005 | 0.964 |
| | 0.825–1.053 | 1.007–1.297 | 1.064–1.962 | 0.780–1.295 | 0.668–1.392 |
| **Education level** | | | | | |
| Illiterate (Ref) | 1 | 1 | 1 | 1 | 1 |
| Elementary education | 1.052** | 1.050* | 0.998 | 1.037 | 1.075 |
| | 1.010–1.097 | 1.003–1.099 | 0.896–1.113 | 0.949–1.133 | 0.950–1.216 |
| High school education | 1.077** | 0.944* | 0.794** | 0.947 | 0.973 |
| | 1.026–1.130 | 0.887–1.004 | 0.690–0.913 | 0.852–1.052 | 0.828–1.143 |
| Higher education | 1.156* | 0.841*** | 0.624*** | 0.966 | 0.905 |
| | 1.003–1.112 | 0.779–0.908 | 0.520–0.748 | 0.859–1.086 | 0.751–1.091 |
| **Currently employed** | | | | | |
| Yes (Ref) | 1 | 1 | 1 | 1 | 1 |
| No | 0.931*** | 1.159*** | 1.426*** | 1.194*** | 1.644*** |
| | 0.904–0.959 | 1.106–1.215 | 1.290–1.577 | 1.121–1.272 | 1.479–1.827 |
| **Household income per capita** | | | | | |
| ≤ half minimum wage (Ref) | 1 | 1 | 1 | 1 | 1 |
| > half but ≤ 1 minimum wage | 1.041** | 1.030 | 1.002 | 1.115* | 1.065 |

(*Continued*)

**Table 5.** (Continued)

|  | Overweight | Hypertension | Diabetes | High Cholesterol | CVD |
|---|---|---|---|---|---|
|  | APR | APR | APR | APR | APR |
|  | 95%CI | 95%CI | 95%CI | 95%CI | 95%CI |
|  | 1.012–1.070 | 0.974–1.089 | 0.904–1.111 | 1.021–1.217 | 0.936–1.212 |
| > 1 but ≤ 2 minimum wage | 1.065*** | 0.989 | 0.955 | 1.188*** | 0.970 |
|  | 1.031–1.100 | 0.938–1.042 | 0.852–1.070 | 1.089–1.295 | 0.843–1.117 |
| > 2 minimum wage | 1.035 | 0.975 | 1.040 | 1.309*** | 0.973 |
|  | 0.998–1.074 | 0.912–1.042 | 0.891–1.214 | 1.173–1.462 | 0.820–1.155 |
| Number of observations | 89,933 | 88,714 | 84,052 | 82,860 | 90,824 |

Source: Brazilian National Health Survey *Pesquisa Nacional de Saúde* (PNS) 2019. APR–Adjusted prevalence ratio; 95%CI– 95% Confidence interval; Ref–Reference category

*p<0.05

**p<0.01

***p<0.001

CVD–doctor-diagnosed with either heart disease or stroke.

age, sex, income, education and employment are more important determinants of these risk factors than household location. Other studies show similar results, including from the *favela* Pau da Lima in Salvador which reported the prevalence of diagnosed hypertension or high cholesterol was no different between the urban slum and non-slum [17]. Evidence from some Kenyan slums showed that the majority of the population had low CVD risk [23], whilst in Haiti, slum populations had a low prevalence of obesity and diabetes [64]. One explanation could relate to survey questions asking if conditions were doctor diagnosed, and socio-cultural attitudes may reduce healthcare use and diagnosis, resulting in under-estimation of the true prevalence in slum and rural populations [10,11,65,66]. For example, if community members are of similarly poor health, risk factors and symptoms can become the norm making atypical symptoms hard to recognise–particularly relevant in slums with overcrowding and increased environmental risk factors [64–67]. Precarious employment, which is likely associated with slum residence, can also inhibit healthcare seeking, even if direct costs of healthcare are low as in Brazil [68,69]. In Brazil, barriers to accessing high-quality care remain and are associated with SES, affecting the likelihood of diagnoses [70]. This likely explains the lower prevalence of metabolic risk factors or CVD in rural populations. For example, accessibility and quality of care are likely correlated with neighbourhood factors such as violence, poverty, transport, geographical remoteness, and other local services [67,71]. Public health services in Brazil, particularly in rural or poor urban areas, are often short-staffed, reducing quality and can result in misdiagnoses and inadequate treatment [66,72]. These factors may also explain some of the differences found by region.

This study found strong socioeconomic patterning across other characteristics, in line with other evidence. For example, increasing education and income were generally associated with lower CVD behavioural risk factors, and a lower likelihood of being hypertensive or diabetic. This patterning is generally found in other contexts and countries [12,41,57,73,74]. Men were more likely to smoke, have diets low in fruits and vegetables, and be physically active–consistent with other studies in Brazil [40]. Other studies also find individuals identifying as of Black or *Pardo*/mixed ethnicity/skin colour have higher prevalence of the behavioural risk factors, hypertension and obesity [41]. This likely stems from additional accumulation of social disadvantage from institutional and structural racism in Brazil [74,75].

This study has limitations. First, the data is self-reported and subject to recall and reporting biases. Second, the cross-sectional nature of this study precludes causal interpretation of the associations found and unmeasured factors could potentially explain the findings. Third, the identification of slum inhabitants involved adapting survey questions resulting in potential for misclassification. While no validation studies of this approach for identifying slum inhabitants have been carried out, we used the UN-Habitat definition, which is widely accepted and has been used in previously published work for assessing slum populations and their health [13,49,50,76,77]. Fourth, the reliance on "doctor-diagnosed" to ascertain health conditions likely does not capture true prevalence–especially for lower SES individuals. Fifth, important CVD risk factors were not included in the survey such as a family history of CVDs, and there is scope for further research on other risk factors. Six, some of the survey questions only captured broad measures of food consumption or risk factors, such as only asking the number of alcoholic drinks consumed and not the alcohol type or only the number of study years and not the quality of education received.

There are policy and clinical implications for this work. People living in slums are more likely to be Black or *Pardo*, of lower income, and with lower levels of education. With their poorer health outcomes, they should be better targeted and monitored by health services. Higher rates of some behavioural risk factors, not explained by SES, in slum populations should be a target of public health and clinical interventions that, if properly tailored, can have a positive impact. Existing programmes such as supervised physical training through the 'Carioca Academy Program' in Rio de Janeiro have already shown to reduce CVD risk [78]. Policies such as the National School Meal Programme in Brazil have resulted in greater access to healthier foods, improved nutrition education and the promotion of local family farming [79,80]. There is also a potentially higher burden of undiagnosed CVD and metabolic risk factors in slum populations. Lack of data on informal urban settlements and the health profiles of these populations means local healthcare services can be poorly targeted, inadequate or culturally inappropriate [81]. Further research should better investigate the specific factors driving poorer behavioural risk factors in slums and also wider inequalities. It is important to understand why greater behavioural risk factors do not translate into higher rates of diagnosed metabolic risk factors within slums, including identifying the roles of health services and environmental factors. Lastly, there should be attention and research involving individuals in non-permanent housing to better understand slum health and the health of the most deprived, vulnerable populations.

## Conclusion

Individuals living in slum populations have higher rates of behavioural CVD risk factors, but similar rates of doctor-diagnosed metabolic risk factors and CVD to urban non-slum populations. Slum environments are likely risk factors for poorer behavioural health outcomes. Large inequalities in both behavioural and metabolic risk factors exist across other socioeconomic characteristics in Brazil. Tackling socioeconomic inequalities in CVD risk factors remains a priority including addressing the factors driving poorer health behaviours in slum populations.

## Supporting information

**S1 Fig. Figures on post-regression predicted prevalence of risk factors by region and area of residence (urban slum, urban non-slum, rural).**
(DOCX)

**S1 Table. Unadjusted regression models on the association between slums and behavioural cardiovascular disease risk factors.**
(DOCX)

**S2 Table. Unadjusted regression models on the association between slums and metabolic cardiovascular disease risk factors.**
(DOCX)

**S3 Table. STROBE checklist for cross-sectional studies.**
(DOC)

## Acknowledgments

JC would like to thank Dr Oyinlola Oyebode, Mrs Marylois Beard, Miss Aditi Rajgopal, Miss Mia Wong, Mr Shinya Monno, and Mr Sam Wong for all their help, assistance and encouragement.

## Author Contributions

**Conceptualization:** Jasper J. L. Chan, Linh Tran-Nhu, Charlie F. M. Pitcairn, Thomas V. Hone.

**Data curation:** Thomas V. Hone.

**Formal analysis:** Jasper J. L. Chan, Anthony A. Laverty, Thomas V. Hone.

**Investigation:** Jasper J. L. Chan.

**Methodology:** Jasper J. L. Chan, Charlie F. M. Pitcairn, Thomas V. Hone.

**Project administration:** Jasper J. L. Chan, Thomas V. Hone.

**Supervision:** Thomas V. Hone.

**Writing – original draft:** Jasper J. L. Chan, Linh Tran-Nhu.

**Writing – review & editing:** Jasper J. L. Chan, Linh Tran-Nhu, Charlie F. M. Pitcairn, Anthony A. Laverty, Matías Mrejen, Julia M. Pescarini, Thomas V. Hone.

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
