## [Decision Letter · Decision Letter 0]

11 May 2022

PGPH-D-22-00446

Inequalities in the prevalence of cardiovascular disease risk factors in Brazilian slum populations: A cross sectional study

Dear Dr.Chan,

Thank you for submitting your manuscript to PLOS Global Public Health. After careful consideration, we feel that it has merit but does not fully meet PLOS Global Public Health’s publication criteria as it currently stands. Therefore, we invite you to submit a revised version of the manuscript that addresses the points raised during the review process.

We look forward to receiving your revised manuscript.

Kind regards,

Peter Bai James, PhD

Academic Editor

Journal Requirements:

1. Please insert an Ethics Statement at the beginning of your Methods section, under a subheading 'Ethics Statement'. It must include:

- (for human participants/donors) - A statement that formal consent was obtained (must state whether verbal/written) OR the reason consent was not obtained (e.g. anonymity). NOTE: If child participants, the statement must declare that formal consent was obtained from the parent/guardian.]

2. Please provide an Author Summary. This should appear in your manuscript between the Abstract (if applicable) and the Introduction, and should be 150–200 words long. The aim should be to make your findings accessible to a wide audience that includes both scientists and non-scientists. Sample summaries can be found on our website under Submission Guidelines: 

https://journals.plos.org/globalpublichealth/s/submission-guidelines#loc-parts-of-a-submission

Additional Editor Comments (if provided):

Reviewers' comments:

Reviewer's Responses to Questions

**Comments to the Author**

1. Does this manuscript meet PLOS Global Public Health’s publication criteria? Is the manuscript technically sound, and do the data support the conclusions? The manuscript must describe methodologically and ethically rigorous research with conclusions that are appropriately drawn based on the data presented.

Reviewer #1: Yes

Reviewer #2: Yes

2. Has the statistical analysis been performed appropriately and rigorously?

Reviewer #1: No

Reviewer #2: Yes

3. Have the authors made all data underlying the findings in their manuscript fully available (please refer to the Data Availability Statement at the start of the manuscript PDF file)?

Reviewer #1: Yes

Reviewer #2: Yes

4. Is the manuscript presented in an intelligible fashion and written in standard English?

Reviewer #1: Yes

Reviewer #2: Yes

5. Review Comments to the Author

Reviewer #1: In this manuscript Chan et al., present the results of their analysis of inequalities in the prevalence of self-reported cardiovascular disease risk factors (CVRFs, both behavioral and doctor-diagnosed) attributable to demographic and socioeconomic factors, specifically examining the inequalities between slum and non-slum inhabitants, using for this purpose publicly available data from the 2019 Brazilian National Health Survey (BNHS). This secondary analysis is innovative in documenting the burden, distribution, and clustering of CVRFs among marginalized populations studied in a well-designed and conducted nationwide representative survey, whose results have important implications for public health policy in the Latin American region, considering the large fraction of population currently living in the precarious conditions that characterize slums.

The main exposure of the analysis is residence as compared to non-residence in a slum, both levels corresponding to urban settings; however, the data set also provides information about rural dwellers which was incorporated into the analysis and reported in the results’ section but without further discussion in the corresponding section. If there was no hypothesis behind rural residency as an exposure, what was the reason for keeping it in the analysis? I am not suggesting that rural residency is not an interesting and important exposure, in fact, rural-urban inequities in CVFRs are rampant in Latin America and your results support this point; however, considering the focus of your analysis, I suggest either developing the discussion about the findings on the rural-urban contrast or excluding rural residency from the exposure at all. The latest can be done by creating a subpopulation within the data set that contains the levels "slum" and "non-slum" (both urban) and conduct the analysis in that subset without excluding rural dwellers.

From the statistical point of view, in estimating prevalence the authors correctly accounted for the sampling design of the survey by incorporating sampling weights (and presumably, clustering); however, at the time of estimating associations, it is not clear to me why they opted for the odds ratios (from logistic regression) instead of prevalence ratios (from, for instance, binomial regression) considering that the first measure of association overestimates the magnitude of relationships in scenarios where outcomes are highly prevalent, as is the case for most of the CVRFs. Also, prevalence ratios are more “natural” estimates of what it is being studied from a cross-sectional design and easier to communicate than odds ratios to the attempted audience of the paper. In consequence, I strongly recommend estimating and presenting prevalence ratios instead of odds ratios.

The discussion, in what refers to the "slum" versus "non-slum" residency contrast, is focused on the main findings of the analysis and framed in the most relevant evidence, pondering the results in light of the limitations of a cross-sectional design and self-reported data on exposures and health outcomes; however, I do not consider the use of regression models a limitation but a strength of the analysis (see lines 388-391). Finally, the authors did an excellent job at developing both policy and clinical implications for their findings.

Minor comments.

In the “Data analysis” section consider making explicit how the complex sampling design of the survey was incorporated into the analysis (i.e., clustering), as well as which variables were included into the adjusted models (only stated in line 389).

In the “Results” section, line 230, perhaps the authors meant to say, “Over two thirds…”.

Reviewer #2: This manuscript assessed data from Brazil's most recent health demographic survey to appraise the association between cardiovascular risk factors and housing conditions. The theme is relevant to global public health; the study is well organized and reads smoothly. This reviewer agrees with the main methodological options adopted here, with special mention to the choice of the database and the statistical analysis. Also, the robust assertion about study limitations is a strong point.

I also underscore the original use of the survey’s data to classify slum households according to UN-Habitat criteria as a study’s strength, which can instruct further research on the subject.

Aiming to contribute to the authors and improve even further this excellent manuscript, I offer two discretionary suggestions.

1. I missed the report on unadjusted odds ratios as supplemental tables. I understand the difficulty of reporting too many regression models. The study collates eleven outcome variables and seven factors. However, socioeconomic factors such as education, employment status, and income correlate among themselves. The problem increases when we consider that race/ethnicity and slum residence can also be associated with the remaining socioeconomic covariates. Excessive autocorrelation can interfere with regression models’ goodness-of-fit indicators. Assessing variance inflation, direct acyclic graphs, and comparing unadjusted and adjusted regression results are tools to attenuate this problem.

2. Brazil is a large country with substantial geographic inequalities. I wonder whether the results reported here would hold for the poorer North and Northeast regions to the same extent as the remaining, wealthier regions. The database is sufficiently large and allows this assessment. Maybe the authors will not have difficulties running the same analytical routines for each geographic region and reporting the results in supplemental tables if the results are eventually relevant.

6. PLOS authors have the option to publish the peer review history of their article (what does this mean?). If published, this will include your full peer review and any attached files.

**Do you want your identity to be public for this peer review?** For information about this choice, including consent withdrawal, please see our Privacy Policy.

Reviewer #1: **Yes: **Victor Herrera

Reviewer #2: **Yes: **Jose Leopoldo Ferreira Antunes

---

## [Decision Letter · Decision Letter 1]

18 Jul 2022

PGPH-D-22-00446R1

Inequalities in the prevalence of cardiovascular disease risk factors in Brazilian slum populations: A cross sectional study

Dear Dr. Chan,

Thank you for submitting your manuscript to PLOS Global Public Health. After careful consideration, we feel that it has merit but does not fully meet PLOS Global Public Health’s publication criteria as it currently stands. Therefore, we invite you to submit a revised version of the manuscript that addresses the points raised during the review process.

We look forward to receiving your revised manuscript.

Kind regards,

Peter Bai James, PhD

Academic Editor

Journal Requirements:

Additional Editor Comments (if provided):

Reviewers' comments:

Reviewer's Responses to Questions

**Comments to the Author**

1. If the authors have adequately addressed your comments raised in a previous round of review and you feel that this manuscript is now acceptable for publication, you may indicate that here to bypass the “Comments to the Author” section, enter your conflict of interest statement in the “Confidential to Editor” section, and submit your "Accept" recommendation.

Reviewer #2: All comments have been addressed

Reviewer #3: (No Response)

2. Does this manuscript meet PLOS Global Public Health’s publication criteria? Is the manuscript technically sound, and do the data support the conclusions? The manuscript must describe methodologically and ethically rigorous research with conclusions that are appropriately drawn based on the data presented.

Reviewer #2: Yes

Reviewer #3: Yes

3. Has the statistical analysis been performed appropriately and rigorously?

Reviewer #2: Yes

Reviewer #3: Yes

4. Have the authors made all data underlying the findings in their manuscript fully available (please refer to the Data Availability Statement at the start of the manuscript PDF file)?

Reviewer #2: Yes

Reviewer #3: (No Response)

5. Is the manuscript presented in an intelligible fashion and written in standard English?

Reviewer #2: Yes

Reviewer #3: Yes

6. Review Comments to the Author

Reviewer #2: (No Response)

Reviewer #3: This is an interesting and well-written manuscript about the relationship between residence in a slum and two forms of CVD risk factors (behavioral and doctor-diagnosed conditions) from the 2019 PNS. While the paper is clear and thorough, there are a few ways it may be further strengthened:

1. Has this approach to defining a slum resident been validated in prior studies (not the definition persay, but this retrospective attempt to define urban residents as slum-dwellers). in the limitations, the authors alluded to prior studies that did this but would be even more convincing if they said how well this approach performed previously.

2. Additionally, the the derivation of the four of the metabolic risk factors was a little bit unclear. Was BMI measured in the survey and used to define overweight? Were the "doctor diagnosed" conditions only self-report (the limitations indicates this is likely the case)? Though mentioned in the limitations, this seems pretty important. Were there biomarkers measured in the survey that could further bolster these definitions or be used as a second way to categorize people as having these risk factors?

3. Table 3 might benefit from the addition of a statistical comparison test.

4. The inclusion of the rural population as a comparison is interesting but sometimes may distract from the focus on slum residents. I wonder if the authors should have made the findings for rural residents (as a comparison) more explicit in the paper and especially in the discussion.

7. PLOS authors have the option to publish the peer review history of their article (what does this mean?). If published, this will include your full peer review and any attached files.

**Do you want your identity to be public for this peer review?** For information about this choice, including consent withdrawal, please see our Privacy Policy.

Reviewer #2: **Yes: **Jose Leopoldo Ferreira Antunes

Reviewer #3: No

---

## [Editor Report · Decision Letter 2]

29 Jul 2022

PGPH-D-22-00446R2

Inequalities in the prevalence of cardiovascular disease risk factors in Brazilian slum populations: A cross sectional study

Dear Dr. Chan,

Thank you for submitting your manuscript to PLOS Global Public Health. After careful consideration, we feel that it has merit but does not fully meet PLOS Global Public Health’s publication criteria as it currently stands. Therefore, we invite you to submit a revised version of the manuscript that addresses the points raised during the review process.

We look forward to receiving your revised manuscript.

Kind regards,

Peter Bai James, PhD

Academic Editor

Journal Requirements:

Additional Editor Comments (if provided):

Please attach a supplementary file that shows that your study adheres to the STROBE guideline for cross-sectional study. You can access a copy of the STROBE checklist for cross-sectional studies via this link below.  

STROBE - Strengthening the reporting of observational studies in epidemiology (strobe-statement.org) 
---

## [Editor Report · Decision Letter 3]

9 Aug 2022

Inequalities in the prevalence of cardiovascular disease risk factors in Brazilian slum populations: A cross-sectional study

PGPH-D-22-00446R3

Dear Chan,

We are pleased to inform you that your manuscript 'Inequalities in the prevalence of cardiovascular disease risk factors in Brazilian slum populations: A cross-sectional study' has been provisionally accepted for publication in PLOS Global Public Health.

Best regards,

Peter Bai James, PhD

Academic Editor